# Attitudes toward War and Peace in the Ukrainian Evangelical Context

**Peter Penner**

Akademie für Kirche und Gesellschaft, 1100 Vienna, Austria; pfpenner@gmail.com

**Abstract:** This article employs the LIM method, complemented by publications and interviews conducted during the ongoing war in Ukraine, to explore the shifts in attitudes toward war and peace within Ukrainian evangelical communities. This shift involves a transition from a pacifist mindset previously predominant among Ukrainian evangelicals to questions about their responsibility and involvement in Ukraine's state and society amid the war that Russia has launched against Ukraine. Interviews with leaders and active church members hint at a possible alignment with Stassen's alternative model of transformative initiatives that might provide potential guidance. While reconciliation initiatives amid the ongoing war may be premature, the article highlights the role of Christian communities in transformative peacebuilding within Ukraine. It is necessary to address tensions within Ukraine. This also entails aiding war-affected individuals, ensuring care for soldiers and civilians, and confronting power abuse and corruption. Ukraine's unity, freedom, peace, and reconciliation must include diverse political and social groups. The article recommends that Ukrainian evangelicals embrace a contextual public theology that advocates for peace, justice, and reconciliation. Ukrainian evangelicals, while supporting soldiers engaged in active combat against Russian invaders, are also contemplating strategies for active participation in peacebuilding and post-war reconstruction.

**Keywords:** war and peace; Ukraine; Russia; Baptistic; Mennonite; Anabaptist; pacifism; just war; transformative initiatives; justice; Sermon on the Mount; reconciliation; Soviet evangelicals; communities; pastors; COVID-19; LIM method; Ukrainian evangelicals

## 1. Introduction

Since 2014, the Ukrainian people, including the evangelical community,[1] have experienced profound challenges. A shift seems to be currently unfolding regarding attitudes toward war and peace within the evangelical communities and leadership. These communities were originally rooted in Anabaptist pacifist principles, which gained acceptance during the era of persecution under the Soviet Union. However, increased interaction with international evangelicals, particularly those from North America, has sparked discussions on the validity of the predominantly pacifist stance of evangelicals, especially within traditional Baptist and Pentecostal denominations.

The military conflict incited in 2014, including the annexation of Crimea and Russian military activities in Ukrainian regions like Luhansk and Donetsk, prompted Ukrainian evangelical leaders to critically reevaluate their Soviet and post-Soviet pacifist inheritance. While this process was partially interrupted by the impact of COVID-19 restrictions[2]— where discussions shifted to how COVID-19 had affected the evangelical church and its liturgical practices[3]—the full-scale Russian war against Ukraine in 2022 reignited intense debates among Ukrainian evangelical pastors, leaders, and communities. Often, these discussions took place in the context of strong controversies between Ukrainian and Russian evangelicals.[4]

As mentioned, historically, Baptistic communities in Ukraine have aligned themselves with pacifist positions inspired by the Mennonites, who have been settled in the region

since 1788 (Huebert 1986, p. 15). However, after the Bolshevik Revolution (1917), these communities grappled with the pressure and persecution imposed by the Soviets, leading some to compromise with the new ideological realities while others emigrated from the area. During that period, Evangelical Christians and Baptists faced similar challenges as they were coerced by the Soviets to renounce their pacifist beliefs and peace witness traditions.[5] The impact of the Second World War and the invasion by fascist Germany further shaped the stance of evangelical pacifist groups. Despite the challenges, many evangelicals remained steadfast in their pacifist positions and resisted coerced involvement in the Soviet military (Jantzen 2023, pp. 295–303). A similar predicament has arisen for Baptistic church communities in Ukraine since 2014, as they face the ongoing war incited by Russia against Ukraine (Soloviy 2023, p. 11).

In response to the devastating war, numerous Ukrainians, including men from Baptistic communities, have chosen to leave the country to avoid engagement in armed conflict. Conversely, others have opted to stay and are confronted with the weighty decision of whether to engage in active combat against the invaders. Through a pastoral lens, this article delves into the challenges faced by Ukrainian Baptistic communities in navigating their role in times of war. By understanding historical contexts, examining responses to previous conflicts, and acknowledging the traumatic current realities of the full-scale ongoing war, this study seeks to provide useful insights for pastoral leaders guiding their congregations through these tumultuous times.

*Proposed Methodology*

The article will adopt the LIM (Loyola Institute for Ministry) methodology developed by Michael A. Cowan to analyze the situation and the changing attitudes toward pacifism and participation in war. This methodology proves particularly useful as it not only helps to provide an overview of the past and present circumstances but also allows for space to offer potential solutions and an outlook for future discussions.

The LIM methodology encompasses four steps for the identification and interpretation of issues (Smith 2016, pp. 205–11). (1) *Identifying a real-life problem*: This will include a presentation of various historical perspectives and influential positions on just war, pacifism, and the involvement of Baptistic communities.[6] Drawing upon instances of pragmatic shifts in pacifist stances from the past, this study aims to highlight analogous historical actions of Mennonite communities, as well as contemporary Baptistic groups in present-day Ukraine. (2) *Interpreting the world as it is*: This step primarily involves discussions that are taking place after the outbreak of a full-scale war in February 2022.[7] The methodology employs a focused qualitative research approach that includes the use of interviews and primary sources such as prominent publications[8] within the Ukrainian Baptistic community (Trofymchuk 2022). (See also the journals of Bogoslov'ki Rozdumy and Bogomysliye[9]). Furthermore, recent contextualized articles from the second edition of the Slavic Bible Commentary (SBC2) and the Central and Eastern European Bible Commentary (CEEBC) will be incorporated as valuable resources.[10] Additionally, relevant articles and discourse published elsewhere in Ukraine between 24 February 2022 and 1 September 2023 will be integrated into the study's source material. Furthermore, pertinent Western literature on war and peace will also be considered. (3) *Interpreting the world as it should be*: In this stage, the research evaluates the present reality using scripture, theology, and Christian ethics, focusing on Glen Stassen's theory of "Transformative Initiatives" as an alternative approach to the traditional "Just War" and "Pacifist" theories (Stassen 2004). (4) *Interpreting our contemporary obligations*: This final stage offers an opportunity to formulate strategic and practical steps, seeking to bridge the disparity between the present circumstances and the envisioned ideal (Tucker 2014, p. 239).

The article will present tentative recommendations and suggestions for fostering positive change, with a focus on Baptistic ministers and active church members, as a possible example for other Christian traditions in promoting peace among the diverse groups in Ukraine. The research will advocate for the adoption of the Anabaptist–Baptist

response by Stassen, showcasing areas where the application of "Transformative Initiatives" can be effective.

## 2. The War in Ukraine as a Real-Life Problem for Baptistic Communities

The full-scale war that Russia started against Ukraine on 24 February 2022 has changed Europe. The war really started already in 2014, but most Europeans have taken notice of it only on 24 February 2022. In the time from 2014 until now, Ukrainian evangelicals seem to have changed their views on the war and on their involvement in it.[11] Before 2014, most evangelicals were supportive of positions close to pacifism (Sannikov 2022a). This is especially true for the Ukrainian Baptists, possibly because they are in historical traditions with the Mennonites.[12] The little group of Mennonite Brethren churches that exists in Ukraine today embraces the theological position of rejecting participation in active combat and, instead, engaging in active aid for civilians and soldiers (Huber 2023). But this is also mostly true for the Pentecostal denominations that build on similar Anabaptist traditions as Baptists (Bremer 2016, p. 14). Other evangelical communities were not as strongly connected to the peace witness tradition.

Mennonites have a long history in Ukraine, dating back to the 18th century when they sought refuge from being forced into military involvement and moved to Ukraine from western and central European countries. Settling in the Russian Empire, they maintained relative isolation from its government, expressing their witness against prevailing cultural norms. Over time, Mennonites prospered in Ukraine, but their convictions were challenged during significant historical events, such as the First and Second World Wars (Bobyleva 2018).

The First World War marked a turning point for the Mennonite community in Ukraine when the Russian Empire demanded their participation in the war. Some Mennonites reluctantly joined the medical corps of the military, while others formed armed self-defense groups to protect their families and villages during the Russian Revolution and the following Civil War (Toews 2013). These events created divisions within the Mennonite community and resulted in the migration of some members to Canada and the United States due to Soviet persecution (Patterson 2020).

The Second World War further deepened the divisions within the Mennonite community in Ukraine. Some were coerced into fighting for the Soviet Army, while others voluntarily or forcibly joined the fascist German army (Goossen 2023, pp. 266–67). Concurrently, Mennonites residing in the US and Canada enlisted in the alliance army to fight against fascist Germany and to openly disassociate themselves from its ideology. These experiences contributed to the emergence of different Mennonite denominations and led some individuals to leave Mennonite churches altogether to join other Christian denominations. During the Soviet Union era, some individuals affiliated themselves with the Baptist or Pentecostal denominations. This fitted well due to historical connections and shared theological beliefs, dating back to the period before the First World War and continuing through the Soviet era, which also included non-involvement in the military or not taking military oaths (Bornovolokov 2022).

Mennonite pacifism and their resistance to participation in war has been extensively studied and serves as a valuable reference for this article's analysis of attitudes in the current war in Ukraine.[13] Studies of Mennonites reveal parallels with the current pragmatic behaviors exhibited by Baptistic communities in Ukraine. The ongoing war appears to significantly impact the perspectives of evangelicals, including Baptist, Pentecostal, and Mennonite Brethren communities, prompting them to reevaluate and contemplate their stance on war. The full-scale war has also had a significant impact on the European continent and has led to a re-evaluation of evangelical perspectives there. The article argues that Christians who encounter any form of war are compelled to critically reexamine their positions, particularly those who are inclined towards pacifist beliefs (Soloviy 2023, p. 11). Historically, this pattern becomes evident in the conduct of Mennonites amidst revolutions and two world wars. Such behavioral shifts and realignment concerning

matters of war, peace, and justice are also discernible within contemporary Ukrainian evangelical communities, particularly among Baptists and Pentecostals, as the subsequent section of this article will demonstrate.

## 3. Interpreting the Current World of Ukrainian Evangelicals in Times of War

This section delineates the empirical study undertaken in the framework of this research. The interviewee group is comprised of individuals from a Baptist background, primarily originating from western Ukraine or having sought refuge in central Europe. It is a convenient sample, as the author had access to these participants for conducting interviews. The outcomes of the study would likely differ if the interviews were conducted with groups from eastern Ukraine, encompassing both the occupied and the regions under Ukraine's territorial control. Respondents from western Ukraine suspected that individuals from eastern regions which are more directly affected by the war would likely exhibit stronger inclinations for active participation in combat.[14]

In an effort to understand the prevailing sentiments and attitudes of Ukrainian Baptistic communities towards war and their potential involvement, the author conducted interviews with a diverse group of individuals. Between June and August 2023, five pastors (aged 40–64), three women who are active church members (aged 46–60), and four young men (aged 19–23) were engaged in a dialogue. Three additional pastors from Ukraine, living and serving presently Ukrainian immigrant communities in central Europe, were interviewed.[15] The total number of participants is 15.[16]

Baptist respondents were deliberately selected from among various Baptistic communities due to their inclination toward a theology of state and church separation. Mennonite Brethren churches, given their limited number and historical presence primarily in the eastern regions, are currently dispersed within non-occupied territory.[17] In the current conflict, their engagement largely revolves around providing humanitarian, medical assistance and chaplaincy. Ukrainian Pentecostal and Charismatic churches exhibit a closer relationship with the state than Baptists. However, they also grapple with similar questions during this wartime context, and the interview outcomes are likely to mirror their respective considerations.

The responses gathered during the interviews revolve around several key themes concerning war and active participation in combat.

### 3.1. Perception of War and a Shift in Pacifist Positions

All interviewees unanimously agreed that engaging in war against the aggressor was the most viable course of action given the current situation in Ukraine. Over half of the participants noted a substantial shift in their previously pacifist position, particularly in response to the Russian invasion of Eastern Ukraine in 2014. Conversely, only a minority expressed opposition to pacifist stances before the invasion. Their comments reflected a common sentiment that, given Russia's invasion of Ukraine, they questioned whether scripture forbids them from defending their fellow believers and relatives. Simultaneously, their understanding of the application of the "just war" theory remained somewhat nebulous; nevertheless, they perceived the defensive actions of the Ukrainian army against the incursion of the Russian forces as morally justifiable.[18]

A significant majority of participants noted that Baptistic churches, pastors, and church members were still in the process of defining a clear stance on war within their respective congregations. Many interviewees (predominantly pastors) shared experiences of dialogues with colleagues from Russia who promoted pacifist ideas while subtly suggesting that Ukrainians should stop resisting the Russian aggression.[19] While many perceived the war as just against the Russian invaders, they did not wholeheartedly endorse either the "just war" theory or pacifism theologically and looked for a third way.[20]

Inquiries regarding the influence of COVID-19 restrictions on evangelical communities' engagement in conversations about the ongoing war since 2014 and on attempts to find an appropriate theological and pastoral response prompted insightful reflections. Participants

unanimously agreed that the imposed restrictions and concentrated discussions on COVID-19 within pastoral and theological circles, as well as among church members, had impeded their ability to engage in theological and pastoral discussions on war and Christian's participation in it.

The prevailing sentiment now among the interviewees indicated a pragmatic approach to the current war situation. Several young Baptist men displayed hesitancy when asked about volunteering as soldiers in combat against the aggressor, even though they acknowledged the imperative to defend Ukraine. From a pastoral perspective, this hesitancy is entirely comprehensible. Instead, they rather were engaged in providing sustenance to soldiers engaged in warfare, offering food, clothing, and equipment. Churches, likewise, displayed this commitment by their concern for individuals within the Ukrainian military, thus exemplifying care and solidarity. A few were actively engaged in tending to the wounded soldiers in adjacent medical facilities.

### 3.2. Perceptions of Government and Military

Simultaneously, the majority of those interviewed conveyed a level of skepticism or mistrust toward the government, also due to their observations of corruption[21] within the Ukrainian military and its inadequate provision of essential supplies like equipment and food to soldiers on the frontline. As a result, many of the respondents took active measures to support the soldiers by providing them with food, clothing, and other forms of assistance, scolding the government's shortcomings in fulfilling these basic necessities.[22]

All interviewees expressed apprehension regarding the potential risk of being forcibly conscripted into the military, which caused young men to adopt discreet behaviors in public spaces to avoid being identified as potential military recruits.[23] Those actively involved in providing support on the front lines mentioned that they strive to keep the young men away from the public eye, in order to prevent them from being drafted into military combat. Particularly, church clergy and women expressed concerns that "their young men might be used as cannon fodder".

Opinions regarding the government's handling of the war varied among the respondents. While some perceived the government to do what is needed, at least half of the participants criticized the authorities for their perceived inability to successfully address corruption and make sound decisions during war. A few drew comparisons between how those with differing opinions were "brought into line", with the prevailing political and ideological interests and the approach of the Russian government which forces its perspective upon others through coercive means.[24] Some raised doubts regarding the religious rhetoric employed by the Ukrainian government, positing that these statements may not necessarily reflect genuine convictions but rather serve as a pragmatic means to communicate with the predominantly Christian populace in Ukraine.[25] The recurring aspiration for a nation characterized by Christian ethics and virtues, which was shared by some evangelical Ukrainians, has been consistently demonstrated to be illusory.

### 3.3. Possible Post-War Developments

Within the interviewed group, a majority was apprehensive regarding possible post-war tensions within society, emphasizing the risk of civil conflict. Some participants called for evangelical leaders to take initiative and contribute to bridging gaps already now, thereby preventing internal societal conflicts. They earnestly requested prayers for evangelical churches, emphasizing their active role and responsibility in sustaining national cohesion and fostering unity among diverse social and political factions. From a pastoral perspective, a significant imperative emerges herein for the ongoing and future education of pastors. Theological educators and pastors in Ukraine would need to facilitate a nuanced position on peace, justice, forgiveness, and reconciliation that transcends Western paradigms and instead prioritizes a distinctly Slavic, contextual, and historically rooted perspective.[26]

All participants concurred that a more sophisticated and nuanced position on war within Baptistic churches would likely emerge after the war concludes. Presently, the responses are characterized by pragmatic reactions, with limited enthusiasm among young evangelical men to volunteer for military service, even though they recognize the imperative to protect Ukraine.

*3.4. Concluding Remarks*

In conclusion, the insights gathered from the interviews provide a partial but still valuable snapshot and perspective on the current state of Ukrainian evangelicals' attitudes toward war. The findings underscore the complexity of the issue and suggest that a more comprehensive and refined theological discourse on war is likely to evolve in its aftermath. The interviews emphasize the importance of justice, forgiveness, and reconciliation as essential components in achieving a comprehensive healing of the Ukrainian nation. This underscores the need for proactive engagement from evangelical entities to actively contribute to this transformative endeavor.

The freedom granted to Ukrainian evangelicals has significantly propelled their development, particularly during the period from 2014 to the outbreak of the full-scale war that Russia started against Ukraine on 24 February 2022. This freedom has not only empowered Ukrainian evangelicals academically but has also fostered increased engagement with society, allowing them to participate more actively in the marketplace. Additionally, this newfound freedom has facilitated dialogue and collaborative academic efforts with historical churches, which traditionally were more open and committed to being actively present in societal affairs.[27] In contrast, Baptist churches adhered to a strict separation of church and state.

As Ukrainian society continues to provide greater opportunities for evangelicals (a broader term that also encompasses Baptist churches), it becomes imperative to sustain conversations not only within the local context but also with Western evangelicals. This dialogue should encompass discussions on how evangelicals perceive their role, whether as influential minorities like in many European contexts, or as politically significant groups like in the USA or the Global South. In their search for a third way of thinking about war and peace—distinct from the just war and pacifist theories—the suggested model, which presents the ethics of the Sermon on the Mount as a framework for "transformative initiatives," stands out as a viable option, rooted in scripture and recognized by theologians within the Baptist communities in Ukraine.

## 4. Exploring the Theory of "Transformative Initiatives"

In this section of the article, following step three of LIM methodology, we delve into theological perspectives to provide a comprehensive analysis beyond the conventional viewpoints adopted by evangelicals. The classical positions of (1) the "just war" (Meisels 2018; Ramsey 2002; Volf and Raychinets 2022, pp. 168–72.) and (2) the "pacifist" (Fiala 2020; Cahill 2019; Kustermans et al. 2019) stance have been extensively examined and differently embraced by various evangelical communities worldwide. The former, often championed by conservative evangelical theologians and churches in the West, finds substantial support in numerous publications (Regan 2013; Reichberg 2017). Conversely, the latter, primarily advocated by groups such as some Baptists, Pentecostals, Mennonites, Quakers, and others, is largely formulated by theologians and churches orienting themselves on Anabaptist positions (L.B. Friesen 2022; Hauerwas 2005).

However, our focus extends to a lesser-known, yet potentially more valuable (3) third theological conviction known as "transformative initiatives" (Stassen 2003c). This perspective emerges from the Christian Ethics group[28] and offers a distinct approach to understanding and addressing conflicts. While scholarly discourse extensively addresses the initial two perspectives, the concept of "transformative initiatives" warrants comprehensive investigation and assessment for its prospective suitability amid the persisting war in Ukraine.[29] By conducting a thorough analysis of this alternative theological paradigm,

our aim is to elucidate its pertinence and potential contribution in cultivating efficacious approaches to navigate the current war. It might possibly delineate a prospective pathway through which notions of justice and reconciliation could be manifested via activities of Ukrainian evangelicals.

The concept of a "transformative initiative of just peacebuilding" finds its foundation in the Sermon on the Mount, with Glen Stassen emerging as its prominent advocate. The theory of "transformative initiatives" diverges from classical pacifism and may not entirely align with the positions held by communities identified as "peace churches".[30] Glen Stassen's influential work, exemplified in his writings, serves as a prime illustration of this approach (Stassen 2003c). In addition to numerous articles on the subject, Stassen's and Gushee's well-known book, *Kingdom Ethics—Following Jesus in a Modern Context*, further elucidates his insights (Gushee and Stassen 2003). The book has not only been translated into various languages but its model has also been explored in diverse contexts.[31] Stassen proposed this model just after the end of the Cold War, and it has since been discussed and tested in different settings.[32] Each context and conflict are unique, and each model requires contextualization and an intimate understanding of local issues, main players, historical and cultural forces. However, Stassen's approach is adaptable and not limited to predominantly Christian contexts. Its appeal, though, lies in its foundation in Christian scripture.

Central to Stassen's analysis is the identification of 14 "triads" within the Sermon on the Mount, denoting distinct text blocks, each characterized by a tripartite structure (Stassen 2003b). Each triad encompasses: (a) a depiction of "traditional righteousness," (b) an analysis of a "vicious circle and its consequences," and (c) a transformative initiative that serves as a pathway to break free from the vicious cycle (Thom 2006, p. 293). It is essential to acknowledge that not all triads directly address the themes of war and peacemaking. Each triad pertains to distinct "transformative initiatives" aimed at introducing justice, healing, and peace through varying approaches. Stassen's paradigm, developed in conjunction with twenty-three other scholars engaged in peace initiatives, highlights the potential for constructive transformative action.[33]

Stassen reveals a prevailing perspective that traditionally regards Jesus' Sermon in the Gospel of Matthew as a discourse on spirituality and piety.[34] However, Stassen emphasizes the paramount importance of recognizing the practical significance inherent in the Sermon on the Mount. Employing the framework of the 14 triads, he draws attention to the existence of "vicious circles" that ensnare individuals and groups in various aspects of life. To overcome such challenges, Jesus presents transformative initiatives as alternative perspectives for addressing a situation. Stassen delves into several specific topics pertinent to the themes of war and reconciliation as emanating from the Sermon on the Mount. The following subsections are derived from Stassen's elucidations:[35]

- "Emphasizing the necessity to forsake all and seek reconciliation with adversaries."
- "Advocating for non-violent direct actions and independent initiatives."
- "Encouraging the inclusion of enemies within the community of "neighbours"."
- "Propounding practices of economic justice, respect for human dignity, and protection of human rights."
- "Acknowledging responsibility for conflicts and injustices while actively pursuing repentance and forgiveness."
- "Promoting the reduction of offensive weapons and halting the arms trade."

Stassen's comprehensive analysis provides profound insights into the Sermon on the Mount's applicability to issues of justice, war, peace and reconciliation. His examination of these specific topics augments the understanding of transformative initiatives that resonate within the context of Jesus' teachings and their relevance in addressing contemporary challenges related to conflict and peacebuilding. While the biblical text of the Sermon on the Mount does not explicitly address military conflicts and their entanglement in various actions and cycles of behavior, Stassen endeavors to move beyond presenting the Sermon as a mere ideal. Instead, he advocates perceiving it as a model, replete with principles and recommendations that possess the transformative potential to reshape the conduct not

only of individuals, groups and church communities and denominations but also of states (Gushee and Stassen 2003, pp. 132–37).

The crux of Stassen's argument revolves around utilizing the Sermon on the Mount's teachings to effect positive change and bring about an end to hostilities while transforming circumstances that might otherwise escalate into conflicts.[36] This active peacemaking stance stands in contrast to pacifism or passive behavior, exemplifying a proactive and engaged approach towards fostering peace and reconciliation. In essence, Stassen's scholarship encourages viewing the Sermon on the Mount as a catalyst for practical, transformative action with far-reaching implications for individuals, groups, churches, and society (McCarthy 2018).

*Evaluating the Model's Applicability in the Context of the War in Ukraine*

The above ideas on "transformative initiatives of just peacebuilding" presented in the context of the United States may appear somewhat idealistic and less applicable to the war situation in Ukraine.[37] There are clearly different societal and political landscapes between the two nations. In the United States, a liberal democracy prevails, and the majority of society has been deeply influenced by Christian ethical foundations, although the state itself adheres to secular principles.[38]

Conversely, Ukraine's democracy is relatively young and thus lacks the maturity observed in established democracies. Furthermore, Ukraine, similar to many Central and Eastern European countries, is transitioning from a long history of totalitarian socialist rule to pursue prosperity and democratic ideals (Szczerba 2022). Due to wartime circumstances, the President of Ukraine has displayed tendencies of messianic and authoritarian leadership, diverging from the Western model of liberal democracy.

Ukraine exhibits a rich and diverse religious landscape, with Christianity pre-dominantly represented by Orthodox, Greek Catholic, and various evangelical denominations, who since 2014 have develop a sense of responsibility for presenting the Christian witness in the public marketplace but have never faced the challenge of doing so during a full-scale brutal war. This results in tensions inside churches and among leaders. On one hand, some still live as "Soviet" or "post-Soviet" evangelicals" adopting a "diasporic", or basement, mentality. On the other hand, there are struggles in terms of how to adapt to the new realities, balancing a sense of belonging and identification with the country of Ukraine, while also remaining committed to the Kingdom of God. As such, a nuanced approach is essential, one that takes into account the unique complexities and dynamics present in the Ukrainian context to develop appropriate and applicable theological perspectives in response to the ongoing war.

The interviews conducted within the framework of this qualitative study reveal a noteworthy shift in the perspectives of the participants. On one hand, there emerges a palpable inclination towards seeking mechanisms to transcend animosity and conflict. These dialogues are characterized by an inherent tension, wherein the participants endeavor to maintain a delicate balance between personal detachment from the war and retaining control over their discourse. Conversely, an earnest yearning for liberty, equity, and instruction within the prevailing circumstances becomes evident. The notions of justice and peace, as elucidated by Stassen, hold a particular allure. A significant proportion of those engaged in these conversations exhibit a proactive commitment to aiding the distressed, the afflicted, and the wounded by the war's impact. Supplying aid, offering prayers and spiritual support for combatants, and safeguarding their well-being are notable examples of this active engagement, which aligns closely with Stassen's concept of "transformative initiatives". It is important to embrace and continue extending and fostering bridge-building, reconciliation, and initiatives that promote peace and justice, breaking free from the cycle of violence and its detrimental consequences. The imperative for robust participation and contribution in the context of the ongoing war calls for the collective mobilization of pastors, active congregants, and diverse church traditions. The overarching

objective being the propagation of peace, freedom, and justice across the variegated strata of Ukrainian society.[39]

## 5. Results and Summary

The use of the LIM method, along with publications and interviews conducted during the war, sheds light on a possible shift experienced by Ukrainian baptistic communities. This shift encompasses the transition from a "Soviet evangelical" and "post-Soviet evangelical" mindset with regard to their relationship with the state and society and leading to important questions regarding the extent to which individuals and communities should be involved and in what manner as the war continues. Are there clear biblical and theological positions of baptistic communities, or at least some guidelines and principles to follow? Some insights shared in interviews by leaders and active church members indicate a tendency that aligns with what Stassen's alternative model of transformative initiatives proposes. While Stassen's model might not provide a roadmap for ending the war, it seems useful in addressing diverse tensions and conflicts prevalent within Ukrainian society and noted within the churches. This article attempts to contribute to the ongoing conversation and exploration of potential pathways.

Articles published in academic evangelical journals since February 2022 engage in a close analysis of the situation and present various viewpoints on how to navigate wartime conditions and respond to them. They offer a valuable diversity in approaches to interpreting the war and addressing the imperative of patriotism, along with fostering a stronger sense of identification between baptistic communities and society. These contextual and relevant wartime considerations are pertinent not only for Ukrainian evangelicals but also for Christianity worldwide. However, there so far exists a dearth of literature addressing the aspects of healing and reconciliation, and the evolving roles of baptistic communities in the aftermath of the war. While the articles in SBC2 offer partial insights, they were written in a quite different context and only partially accommodate the present experiences of Ukrainian evangelicals. A revision and expansion of the SBC2 in the Ukrainian language would clearly amplify its contextual applicability. The revised commentary could then be leveraged as a valuable resource, equipping pastors and churches within Ukraine with a pertinent tool.

The article examines the potential of the "transformative initiatives" model for reconciliation and for just peacemaking by baptistic communities and churches in Ukraine. Presently, achieving such reconciliation seems impossible and premature, given the ongoing war in Ukraine. Nevertheless, the hope for eventual reconciliation remains present among Christian communities, and so the need for Christians to act as instruments of transformative peacebuilding within Ukraine itself is emphasized.[40]

In times of war, the primary focus of the "transformative initiatives of just peacebuilding" and of breaking "vicious circles and its consequences" should be directed towards different groups within Ukraine. Baptistic communities can initiate a process of fostering peaceful and inclusive language practices (Baumeister 2023). Overcoming separation and distrust among churches, unions, and various evangelical groups is crucial. Additionally, building trust and reconciliation with the historical churches in Ukraine, including the Ukrainian Orthodox Church[41], is necessary for fostering unity and peace.[42] The seminars and conferences organized by EETI (https://eeit-edu.info, accessed on 19 September 2023) play a valuable role in fostering connections and facilitating dialogue among diverse Christian traditions.

Addressing the tensions between that exist in churches and society is essential to avoid escalating conflicts within the country. Stassen's "transformative initiatives" present tangible methods for initiating and sustaining dialogue, as well as overcoming barriers between distinct groups in Ukraine. Providing aid to those affected by the war and ensuring proper physical, medical, and spiritual care for soldiers and civilians is an equally crucial aspect of evangelical presence in Ukrainian society. Addressing issues of corruption that tend to be amplified during war is part of speaking truth to power.

In the context of the single nation of Ukraine, unity, freedom, peace, and reconciliation must encompass the diversity of political and social groups. Wise leaders in Baptistic communities might help to overcome those tensions inside Ukraine by bridging the gap inside their own communities.

The article suggests that Ukrainian Baptistic churches and communities embrace a contextual public theology that advocates for peace, justice, truth, forgiveness, and reconciliation. Such approaches will enable them to become impactful witnesses for renewal and reconciliation in the nation. Ukrainian evangelicals, even as they actively reexamine their attitudes on 'just war' and pacifism faced with the ongoing war against Russian invaders, are also deliberating on strategies to actively participate in peacebuilding and the reconstruction of their society and nation. These considerations include preparations for the post-war era.

**Funding:** This research received no external funding.

**Informed Consent Statement:** Informed consent was obtained from all subjects involved in the study.

**Data Availability Statement:** The data presented in this study are available in the article.

**Conflicts of Interest:** The author declares no conflict of interest.

## Notes

[1]　Before the war, the Protestant and evangelical population in Ukraine exceeded 600,000, constituting up to 2% of the total population. Approximately 50% of this demographic would align with the characteristics of baptistic churches. Ukraine, akin to Romania, boasts one of the largest Protestant and evangelical communities in Central and Eastern Europe. The term "evangelical" in the context of this article describes a broad spectrum of Protestant and evangelical church groups and traditions. It does not specifically refer to denominational names such as "Evangelical Christians" but rather encompasses all Protestant-evangelical churches and denominations. The term "baptistic" in this article specifically refers to denominations such as Baptists, Pentecostals, and Mennonites that trace their historical roots to Anabaptism where, among other aspects, a strict separation of church and state was emphasized, see: (McMillan 2021, p. 66).

[2]　Kraliuk et al. (2020). And specifically on Protestants: (Balaklytskyi et al. 2021, p. 162; Komlev 2020).

[3]　Since the outbreak of COVID until the full-scale war in February 2022, a significant amount of attention was devoted to theological debates regarding COVID, resulting in less attention to the ongoing conflict in Eastern Ukraine and other war-related issues, which deserved higher priority.

[4]　This article does not consider Russian evangelicals or the tensions between Ukrainian and Russian baptistic communities. An article focusing on evangelicals in Russia would differ significantly, due to the limited possibilities for open discussions and publications in that context. There is also a great diversity of perspectives among Russian evangelicals, with some tending to adhere to pacifist positions. Certain sections, especially Stassen's "Transformative Initiatives," hold similar value for application within the Russian evangelical context. Reading and contextualizing this article for Russian evangelicals, particularly in light of Mt 7:1-5, is recommended as guidance for Russian evangelicals examining the experiences of their Ukrainian counterparts.

[5]　See especially the subchapter on "Baptists and peace witness" in: (Penner 2007, pp. 188–92).

[6]　Patz (2022). See also other articles from the same volume.

[7]　Early voices on the Russian war against Ukraine: (Geychenko et al. 2022).

[8]　See recent publications in *Theological Reflections*, part of vol. 20,1, vol. 20,2 (2022) and vol. 21,1 (2023) (http://reflections.eeit-edu.info/issue/archive, accessed on 18 August 2023). The Eastern European Institute of Theology (EETI), directed by Roman Soloviy, publishes "Theological Reflections (Богословські Роздуми)," an evangelical scholarly journal. In collaboration with Western evangelical partners, the Institute has orchestrated numerous seminars, often featuring prominent scholars from Western evangelical institutions, to address the topic of war. Within the pages of "Theological Reflections," certain articles mirror these Western perspectives, while others are authored by Ukrainian evangelical scholars.

[9]　The journal "Bogomysliye" holds substantial readership within the evangelical community, particularly among Baptist pastors and theologians: http://almanah.bogomysliye.com, accessed on 18 August 2023.

[10]　Constantineanu and Penner (2022), abbr. CEEBC. (Sannikov 2022b), abbr. SBC2. It is important to note that while both commentaries possess an academic foundation, their primary audience comprises pastors and active church members within the Central and Eastern European and Eurasian regions, respectively. These commentaries, available in English and Russian, respectively, reflect the collective expertise of scholars from this region. Upon initial examination, a comparison of the two commentaries, along with the social and political articles they encompass, demonstrates a shared focus on the subject matter. However, a notable distinction emerges in the extent of this focus; the CEEBC encompasses twice the number of articles on

social and political aspects as compared to the SBC2. The Slavic Bible Commentary's first edition (SBC1), published in 2016, was printed in both Ukraine and Russia. This approach was undertaken due to the logistical challenge of effectively distributing the commentary to Russian-speaking individuals given the already existing hostilities between the two countries. In contrast, the subsequent second edition of the commentary was exclusively printed and disseminated within Ukraine.The SBC2 was produced in a period marked by escalating tensions post-2014 and the early stages of the full-scale war (2022). In contrast, the CEEBC was developed in the aftermath of the Yugoslav wars and during the Russian invasion of Ukraine. A distinguishing feature of both commentaries lies in the firsthand experiences of the authors, many of whom have directly encountered the era of communism and the subsequent transitional periods in this region. As a result, their contributions serve as valuable insights for comprehending the nuanced context that informs these texts.

[11] These observations are drawn from the author's experiences and interactions since his visit to Maidan in 2014, where he engaged in numerous conversations with colleagues from theological seminaries and pastors.

[12] Mennonites had a significant historical presence in Ukraine and contributed to the 19th-century revival in Ukraine and Russia, as well as to the development of the Baptist movement. During the Soviet era, Mennonites, Baptists, Pentecostals, and other religious groups were compelled by the state to form a single union.

[13] L.G. Friesen (2022). Much has been published in Mennonite historical journals, research of archives, many personal narratives and books.

[14] Comments from interviews with pastors.

[15] Numerous individuals have emigrated from Ukraine, including a substantial number of evangelical Christians. A debate ensues regarding the assessment of patriotic allegiance: whether it aligns with those who sought refuge or those who remained within the country. This issue holds particular significance for pastors who have church administrative and pastoral responsibilities, as they grapple with the decision to either stay or leave. Both options for emigration and choosing to remain in Ukraine present distinct challenges and potential avenues for ministry. See (Teteryatnikov 2022) and (Ten 2022).

[16] Since the commencement of the war in 2014, the author has engaged in numerous conversations with Ukrainian evangelical individuals, meeting with over 40 people during his visit to Ukraine just in June 2023. From this larger group, the author judiciously selected representatives for interviews based on insights gained from previous conversations. The chosen 15 individuals effectively capture the diverse perspectives within the broader evangelical community in Ukraine.

[17] See more on their positions in: https://anabaptistworld.org/?s=Ukraine, and also in: https://mcc.org/search?keywords=ukraine, accessed in 18 August 2023.

[18] The article by (Sannikov 2022a) within the SBC2 presents an approach to the concepts of war and pacifism aiming for a neutral stance. However, such neutrality is primarily feasible in periods of peace. In light of the full-scale war, a more explicit and transparent position would better serve the needs of pastors and churches. Additionally, it is worth noting a more contemporary article that grapples with the challenge of establishing a distinct stance amidst the dichotomy between pacifism and militarism: (Fimushkin 2022).

[19] Further insights into perspectives of Russian evangelicals regarding the war and the aggression perpetuated by the Russian army against Ukraine can be gleaned from the following article: (Nesterenko 2022). Ukrainian authors have also challenged responses of Russian popular pastors and leaders, urging them to reconsider their stances on the war. A notable instance of this is exemplified by (Kravtsev 2022), https://gazeta.mirt.ru/stat-i/cerkov/post-2558, accessed 18 August 2023, or (Kravtsev 2023), https://xmegapolis.com/xtianity-zeal-0706-2023, accessed 18 August 2023.

[20] The responses reflect a nuanced pattern, acknowledging the presence of complex realities that question the direct application of classical pacifist stances. Engagement with Western evangelical perspectives advocating a 'just war' approach has prompted local churches to reassess their traditional positions. In contrast, Tkachenko's discourse presents insightful observations derived from his exploration of the "just war" concept (Tkachenko 2022). These observations shed light on the challenges that Ukrainian evangelicals encounter when attempting to put this theory into practice, as well as when grappling with the principles of the "pacifist" doctrine. This prompts the need for an alternative perspective.

[21] Statements such as "the government neglects corruption during the war because it is afraid to lose supporters in the war against Russian invaders", "many continue to engage in theft in the shadow of war", "nothing has changed: you need something done, don't be surprised to be asked for bribes" were common.

[22] This strongly resembles perspectives on patriotism, even among those who might lean towards pacifist viewpoints, aligning well with the proposed stance of Stassen. Upon contrasting the discussions in CEEBC and SBC2 on patriotism and nationalism, it becomes evident that Dănuț Mănăstireanu's article resonates more directly with the current wartime circumstances in Ukraine (Mănăstireanu 2022, p. 454). He links this concept to the Balkan wars, discusses the role of culture and ethnic diversity as enduring creations of God even beyond the transient world, and highlights the presence of a "quasi-religious dimension of nationalism" that should not be found among Christians. Maintaining a patriotic perspective involves both supporting the people and offering constructive criticism of authorities when their actions are misguided.

[23] The personal challenges faced by young people as well as their parents and churches partially reveal a gap in knowledge and an uncertainty in behavior for which they may not have been adequately prepared. The situation also raises questions about individual decision-making, particularly in the absence of a clear stance on engaging in military action and pacifism. In his article,

Sannikov expounds upon historical instances where Christians have sought alternative avenues beyond direct engagement in armed combat to contribute to the betterment of their nation and fellow citizens (Sannikov 2022a, p. 1305). The article presents valuable insights for pastoral guidance and ecclesial discourse. It also discusses safeguarding of the family unit, protection of vulnerable individuals, and voluntary enlistment for military service, where individuals willingly undergo training and stand prepared to engage in lethal combat against adversaries as directed by wartime governance.

24 Extensive literature has explored the concept "Russian world," while only sporadically suggestions that a potential parallel notion, the "Ukrainian world," might exist have been voiced. Further insight into the "Russian world" can be found, among others, in: (Shishkov 2022).

25 The article authored by Fedor Raychinets on "Church and State" holds particular relevance in this context (Raychinets 2022). However, the inclusion of further contextualization specific to Ukraine's historical and contemporary situation would be immensely beneficial. Insights available within SBC2 and CEEBC, also on the subject of "Bribery and corruption," offer additional valuable perspectives: (Timchenko 2022; Borzási 2022).

26 While many would concur that this is a very important issue, locating relevant literature directly addressing this topic was challenging. It might need to be written yet.

27 Numerous Ukrainian evangelical scholars have benefited from academic scholarship and studies originating from the West, including North America and Europe. The current generation of theologians exhibits a significant reliance (until 2014) on Western academic theological contributions and engagement. Consequently, Stassen's approach has exerted a considerable influence, particularly on those who studied at the International Baptist Theological Seminary and engaged with him personally. Additionally, evangelicals in other Eastern European countries, such as Bulgaria, have translated Stassen's major work (*Kingdom Ethics: Following Jesus in Contemporary Context*), authored reflective articles, or embraced his teachings through supportive arguments.

28 Stassen describes the process and persons involved in the (Stassen 2003a). Discussion results were published in (Stassen 2004). See also: (Anderson and Rector 2014).

29 Parush R. Parushev and Fyodor Raychynets address the beginnings of the war in 2014 in an issue devoted to Stassen: (Parushev and Raychynets 2014). See also: (Stassen et al. 2013). Recent publication on "breaking cycles of violence": (McCarthy 2020). Different possible scenarios can be found and are proposed in: (Stahn et al. 2020).

30 These are Christian traditions such as Mennonites, Quakers, and others.

31 The *Journal of the Society of Christian Ethics* (23/1 in Spring/Summer 2003), for example, provides case studies of the model being used, with varying success, in post-colonial Africa, specifically Liberia and Rwanda, in the Middle East, and in humanitarian interventions.

32 See the Special Issue of *Christian Ethics Today* 22,4, aggr. iss. 95 in Fall 2014 titled "The Global Relevance of Glen H. Stassen and Just Peacemaking: Essays by His Friends in Various International Settings".

33 The results of this discussion are also available under the title: "Just Peacemaking: Ten Practices for Abolishing war" Pilgrim Press: https://d3n8a8pro7vhmx.cloudfront.net/unitedchurchofchrist/legacy_url/6534/Just-Peacemaking-practices.pdf?1418 431403, accessed 18 August 2023. See the full version in the book: Stassen, Just peacemaking.

34 Jesus addresses "the crowds" and disciples (Mt. 5:1) in the Sermon on the Mount using the second person plural, "you," to engage with the listeners. Matthew presents this text in the Gospel with the intent of addressing the reading church.

35 Sub-chapters in: (Stassen 1992).

36 In the interviews, it was often expressed that tensions already exist but may escalate further once the war is over. Therefore, it is important to engage in preventive work now. Stassen offers strategies to prevent escalations within society.

37 Searle (2022, pp. 187–89). Having resided in Ukraine for a longer time, Searle is well-acquainted with Ukrainian baptistic communities. His endeavor to correlate Stassen's proposal with the perspectives of Orthodox philosophers such as Florovsky, Solovyev, and Berdyaev presents an intriguing concept warranting examination for its pertinence within the ongoing war in Ukraine.

38 Publications and models on peace, justice, and reconciliation coming from the former Yugoslavian context and the wider Central and Eastern Europe might be more applicable: (Constantineanu 2016, pp. 684–91). The interview conducted by Miroslav Volf with Fedor Raychinets—(Volf and Raychinets 2022)—offers a valuable scholarly and pastoral contribution. The focus primarily revolves around topics of war, hatred, and forgiveness, and their implications for both Ukrainian and Russian faith communities. The interview proposes approaches for Ukrainian and Russian evangelicals to navigate the crisis in accordance with their faith and beliefs. The interview also alludes to Volf's latest book, *Life Worth Living* (Volf et al. 2023, p. 163), which delves into his father's wartime experiences. While the interview's insights are particularly pertinent to the Ukrainian context, their application may necessitate an initial focus on the internal dynamics within Ukraine before possibly being extended to Ukrainian-Russian evangelical relations. In its entirety, Volf's interview, along with his published works, serve to contextualize healing within the Ukrainian community and to empower Ukrainian evangelicals to assume a prophetic and reconciliatory role. Miroslav Volf has previously addressed the topics of war, justice, peace, forgiveness, and reconciliation, notably in his major work, (Volf 2019). An earlier contribution is his article, (Volf 1998). While Volf's approach offers valuable insights, the author opted for Stassen as including both scholars would go far beyond the scope of this article.

39 For a comparison of the articles on social, economic, and political issues utilized from SBC2 and CEEBC, please consult footnote 8.

40     The current time necessitates a careful consideration of the diverse guiding principles proposed by figures such as Stassen and Volf as well as Eastern European theologians. This reflection is essential for pastors and members within baptistic churches. Their collaborative efforts are crucial in instigating a timely process focused on achieving justice, peace, and reconciliation within the nation.

41     It is distinct from the Orthodox Church of Ukraine.

42     Ukrainian evangelicals find themselves within a broader religious and cultural landscape, which is predominantly Orthodox and Greek Catholic. These historical churches possess extensive experience and actively engage in discourse surrounding the ongoing war. Cyril Hovorun, among others, is a prominent voice contributing to this dialogue through numerous publications and discussions. While this article acknowledges the value of these discussions and publications, it was not possible to incorporate them due to constraints of this limited research.

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
