# Peer review of "Attitudes toward War and Peace in the Ukrainian Evangelical Context"

_religions, doi:10.3390/rel15010024_

Round 1

Reviewer 1 Report

Comments and Suggestions for Authors

The argument of the article could be enhanced by stipulating how the mainly individual ethics of the Sermon on the Mount can be translated into concrete believers-group actions applying the "transformative initiative" described in the article.

Author Response

Dear Reviewer, I sincerely appreciate the time you dedicated to reading my article and providing valuable feedback to identify shortcomings and areas for improvement. Your insights have been instrumental in enhancing the quality of my work. I have uploaded the revised article, and I trust that it now meets the standards for publication. Thank you once again for revisiting it and considering my responses to your comments.

Reviewer 2 Report

Comments and Suggestions for Authors

Fascinating article. I added some remarks to the text attached. Please take them seriously.

Author Response

(The authors gave the same response as above.)

Reviewer 3 Report

Comments and Suggestions for Authors

Review Report

Paradigm Shift on Attitudes Toward War and Peace in the Ukrainian Evangelical Context

Journal Religions (ISSN 2077-1444)

Manuscript ID religions-2658209

Section Religions and Theologies

Special Issue

Pastoral Theology in a Multi-Crisis Environment

Summary in one paragraph

The article aims to examine the attitudes of Ukrainian evangelicals concerning their support for and active participation in war and highlights the role of Christian communities in transformative peacebuilding within Ukraine.

Using a pastoral lens, it deals with the challenges faced by Ukrainian baptistic communities in navigating their role in times of war and seeks to provide valuable insights for pastoral leaders guiding their congregations through these tumultuous times. It recommends that Ukrainian baptistic churches embrace a contextual and public theology advocating for peace, justice, and reconciliation.

The article fills a gap in dealing with sensitive and controversial topic of healing and reconciliation in post war Ukraine. Current articles do not deal with the various tensions and conflicts within the complex context of Ukrainian society. It provides insight in internal tensions and different positions.

The article seeks to facilitate a "nuanced positioning that transcends Western paradigms, prioritizing instead of distinctly Slavic, contextual and historically rooted standpoint”(pg. 9, footnote 40). The author offers a "nuanced approach... taking into account the unique complexities and dynamics present in the Ukrainian context", in the "transitional state" of current Ukraine, marked by "tensions" and a "struggle" to on how to "balance a sense of belonging and identification with Ukraine, with a commitment to the Kingdom of God. It is all about developing "appropriate and applicable theological perspectives in response to the ongoing conflict" (pg 14)

This article emphasizes the "need for Christians to act as instruments of transformative peacebuilding within Ukraine itself" (pg15). It also focuses on the need to "building trust and reconciliation with the historical churches in Ukraine, including the Russian Orthodox Church, necessary for fostering unity and peace. (pg 15)

The author is in a unique position to address such a sensitive and controversial topic as an insider outsider, taken into consideration his roots and ministerial focus in the context of the former Soviet Union.

General concept comments

Article

For those not familiar with the complex context and background of the Ukrainian ecclesiastical landscape, the use of the various concepts Evangelical, Baptistic, baptistic’, Mennonite is sometimes confusing, sometimes used as synonyms, sometimes as complementary and sometimes as an overarching concept. It might help to add some explanatory notes. See my comment later under the heading “quality”.  

Methodological inaccuracies

There are no methodological inaccuracies. The methodology Loyola Institute for Ministry (LIM) developed by Michael A. Cowan is used to analyse the situation and the changing attitudes. The article focuses on the Anabaptist response by Glen Stassen using his theory of "Transformative Initiative" as an alternative approach to the traditional 'just war" and "pacifist theories" (pg 4) The LIM methodology is used as a valuable attempt to offer a theological foundation to build bridges in complex situation with many internal tensions.

Review

The review topic is well covered. It is well-documented with the most recent publications, and with appropriate references and deals with a wide range of perspectives. The author shows to be deeply embedded in the topic, while keeping a critical distance. He offers a unique pastoral perspective as an insider outsider.  The review topic is of high relevance, although some insiders might consider it as “premature”. The author displays courage to deal with this unique perspective, clearly filling a gap in knowledge, by employing a “western” methodology to analyse the war and peace attitudes in Ukraine and seeking to transcend “western” paradigms by offering a contextual perspective. Empirical research is used as complementary.

With regard to the completeness of the review topic, although reference is made to the complex wider Ukrainian ecclesial context and to an important figure of the Ukrainian Orthodox Church, Cyril Hovorun, no mention is made to his (recent) publications.

Specific comments

inaccuracies

  1. Add Ukrainian evangelical as key word
  2. pg. 3 confusing footnotes 12 and 13 and their placement. Proposal to transfer footnote 12 with reference to Trofymchuck following Ukrainian baptistic community. Rest of footnote 12 could be put behind "prominent publications", and include reference to "see present publications on war, etc... A new footnote 14 could offer explanation to the journal "Bogomyslie".
  3. pg 3 footnote 14: add abbreviation (SBC2) following reference to Slavic Bible Commentary ed. 2. In same footnote add abbreviation (SBC 1) following Slavic Bible Commentary's first edition.
  4. pg 4 footnote 14: "by escalating tensions post-2000", I guess this should be "post-2020".
  5. pg. 4 Footnote 17: It might offer more clarity to the wider context, scope and limitation of this article to transform this footnote into a new paragraph in the main text, in the introduction of this article. In this same paragraph footnote 67 could be transferred to the main text, adding more information on the wider context, and clarifying the delimitation. A reference to some recent publications of Hovorun might be helpful.
  6. pg 6, fourth paragraph 4th sentence. Why adding the adjective "mature" to female active church members, and not to the "four young man"?
  7. pg. 7, heading Critique of Ukrainian Military slightly overlaps with the paragraph on lack of confidence in governance.
  8. pg. 10, footnote 40. Some German sentences need to be taken out. However, I agree with the contents of this comment, that it might be fruitful to transfer part of this footnote to the main text.
  9. pg. 10, paragraph 3. It seems that this sentence is meant to be a new heading.
  10. Pg. 10, paragraph 5. It might also be helpful to add a new heading: "Conclusion", as it seems that this paragraph starting with "in conclusion" is related to all the elements of the empirical research.  This paragraph could even be transferred to the section with the conclusion of the whole article.
  11. 0pg. 12 1st paragraph. I propose to take out the words "into Ukrainian Society" as this paragraph is a presentation of Stassen's "triads". The connection to the Ukrainian context is explained later on pg. 13, under the heading "Evaluation the applicability..." 
  12. pg. 15 footnote 68: add date of access.

unclear sentences

1.      pg. 9 2nd paragraph: add connecting word like "Therefore", following sentence ending with "comprehensible".

2.      pg. 11 2nd paragraph. It is unclear what is meant with the "Christian Ethics group". Reference needed.

Rating the Manuscript

Novelty:

The Research question is original and well defined. The results provide an advancement of the current knowledge.

Scope:

The article fits the journals scope, and the scope of this thematic issue. With a pastoral approach, an attempt is made to offer theological perspectives to initiate conversation on building bridges between dominant paradigms in the Ukrainian evangelical churches.

Significance:

The results are interpreted appropriately. The article is of great significance, offering a new perspective to a very complex situation and offering unique insights in the complexities and background of the ecclesial and political situation.

The article fills a gap in dealing with a very sensitive and controversial topic of healing and reconciliation in post war Ukraine. Current articles do not deal with the various tensions and conflicts within Ukrainian society.

Quality:

As mentioned earlier, to avoid confusion to outsiders to the Ukrainian church situation, it might be helpful to revisit and clarify the use of key concepts of evangelical, 'baptistic’, baptistic, Baptist, Mennonite, by adding more consistency and some explanatory notes. Sometimes they are used as synonyms, sometimes as complementary concepts, sometimes one used as overarching concept.

Scientific soundness:

A careful use is made of a wide range of sources. Empirical research is used in a complementary way. The article provides unique and very important insights and background information into a very complex context.

Interest:

The article is comprehensive, and likely to draw a wide readership, as it deals with an important and relevant topic in a pastoral, not offensive tone, from a unique insider outsiders’ perspective. For sure it will stimulate conversation and discussion into what is considered by some as a controversial topic. It requires someone like the current author to raise these issues with integrity, credibility, and seniority.

Overall merit:

There is surely an overall benefit to publishing article, as it deals with a highly relevant topic on war and peace in Ukraine, from a unique perspective.

Overall Recommendation:

Accept after Minor Revisions. Some minor improvements of inaccuracies are needed. 

Comments on the Quality of English Language

The quality of English is good.  The article reads well. Some minor proposals to improve readability and correct inaccuracies are included in the previous section. 

Author Response

(The authors gave the same response as above.)

Reviewer 4 Report

Comments and Suggestions for Authors

The article "Paradigm Shift in Attitudes Toward War and Peace in the Ukrainian Evangelical Context" appears to lack methodological substantiation. The author fails to clarify the meaning of "paradigm shift," including why and how it occurred in these specific historical circumstances. The most intriguing part of the article is the section reflecting on interviews conducted with Ukrainian evangelicals. However, the discussion of these interviews reveals the author's fundamental misunderstanding of how to effectively conduct qualitative research. Relying on just 15 interviews, the author reports percentages of participants holding certain views without acknowledging that a sample of 15 cannot represent the entire community. The value of the interviews lies in their narrative construction, not in the number of people sharing specific opinions.

The author suggests applying Glen Stassen’s concept of "transformative initiative of just peace building" in Ukraine by Ukrainian evangelicals. However, this suggestion seems irrelevant and unfit to address the Ukrainian situation. Although the author acknowledges that Stassen's concept, originally presented in the context of the United States, might seem “idealistic and less applicable” to Ukraine's war situation, the outlined reasons come across as irrelevant and demeaning towards Ukraine. The author contrasts the US as a "liberal democracy" deeply “influenced by Christian ethical foundations” with Ukraine, portrayed as fundamentally different. It would be beneficial for the author to adopt a broader perspective, recognizing that Ukraine is defending itself against one of the world's largest authoritarian and autocratic countries, whose military has committed an unprecedented number of war crimes, affecting millions of civilians, including children.

Moreover, the author should consider that Stassen's concepts were likely not designed for a country suffering a brutal military invasion from a much larger and more powerful neighbor, as Ukraine is. This significant context seems entirely disregarded in the author's conclusions and analysis. Discussing the interviews, the author overlooks the possibility that participants may be, and most likely are, experiencing profound trauma from the war's unprecedented cruelty and injustice against Ukrainians. The author claims that Ukrainian Evangelicals "support the conflict," despite the fact that all Ukrainian churches, including Evangelicals, have condemned the war and Russian invasion from the start. Such statements by the author are deeply unjust and hurtful toward Ukrainian Evangelicals and Ukraine as a whole.

In summary, the article reflects the author's biased view of the Ukrainian situation, which prevents them from accurately handling the research material and reaching convincing conclusions. The article needs significant revision.

Comments on the Quality of English Language

It sounds like the text was translated into English by AI and appropriate proofreading was missed

Author Response

(The authors gave the same response as above.)

Round 2

Reviewer 4 Report

Comments and Suggestions for Authors

The revised version of the manuscript is better than the initial version. However, the author still fails to develop a solid argument for their main thesis, as stated in the abstract, particularly regarding the importance of implementing 'transformative peacebuilding' in Ukraine. The abstract mentions 'addressing tensions within Ukraine – between Eastern and Western regions, and between citizens and economic elites.' Although the article briefly discusses some respondents' complaints about the Ukrainian government and current politics, which are typical in democratic countries where citizens often critique their elected officials, it does not address the mentioned 'tensions' between the Eastern and Western regions of Ukraine. The author does not clarify what these tensions are or why they should be considered as such. Making statements in the introduction without further explanation in the main body of the article seems unreasonable.

On page 7 of the revised manuscript, the author mentions 'encompassing both the occupied and the regions still under Ukraine's territorial control.' The use of 'still' here is ambiguous. Does it imply that the author expects further Ukrainian territories to be occupied by Russian forces? Or is the author anticipating this to validate their concept? Clarification is needed in this context.

The author's evident emotional desire to foster reconciliation among Ukrainians, 'including those often perceived as collaborators and pro-Russian' (page 20 of the manuscript), is poignant. However, it would be prudent for the author to consider that collaborators with Russian occupational administrations are typically prosecuted according to the law, as they are often involved in acts of violence against the Ukrainian people, leading to numerous deaths and casualties. I suggest that the author should more thoroughly explore the situation and gain a deeper understanding of its social and legal aspects before advising Ukrainians on how to treat collaborators.

Author Response

I appreciate your thorough review of my manuscript for a second time. In response to your comments, both in my written responses and within the manuscript itself, I have worked on addressing the various issues you raised. I trust that the changes made will meet your expectations and contribute to the successful publication of this material. For your convenience, I have attached the form containing my responses and kindly request you to review the revised manuscript.
